# Three-Point Inverse and Forward Kinematic Algorithms for Circle Measurement from Distributed Displacement Sensor Network

**DOI:** 10.3390/s19214679

**Published:** 2019-10-28

**Authors:** Mohammad Mayyas

**Affiliations:** eFactory Laboratory, Mechatronics Engineering Technology, Bowling Green State University, Bowling Green, OH 43402, USA; mmayyas@bgsu.edu

**Keywords:** distributed sensors, arc fitting, roundness, arc radius inspection, manufacturing metrology, TPFK algorithm, TPIK algorithm

## Abstract

Automatic fitting of an arc center and radius is a quality problem frequently encountered when manufacturing a mechanical component. Due to the complexity of the measurement, validating each manufactured component via inspection is not feasible or economical. This paper introduces a new validation procedure for measuring arcs from distributed sensors. The goal of this proposed measurement process is to improve measurement throughput (i.e., parts measured per unit of time) and reduce measurement errors associated with hardware and algorithms. This proposed model develops a three-point inverse kinematic algorithm (TPIK) accompanied by a calibration master to obtain the relative location of the measurement system by solving a set of six non-linear equations. This technique allows deployment of a high accuracy gauge systems that in general, reduces machine and algorithm errors. The direct fitting is validated by using mathematical, CAD, and experimental models. Furthermore, a modified definition for the roundness index is introduced based on the proposed forward and inverse algorithms. The simulations examine the roundness index in relation to the measurement precision, sampling angle, nominal radius, and part variation. A benefit of this proposed method is accurate and rapid inspection of the radii and elimination of the human error associated with part loading variation during conventional radii measurement. The rapid, accurate inspection and corresponding reduction in human error make this method an excellent process for inspection of large quantities of components.

## 1. Introduction

Fitting surface geometry to a part is the process of estimating the interpolated geometry between the real data points collected using metrology instruments on real components. This method is an efficient way to develop a geometric model of the feature. The problem of fitting a circular arc (say, for a fillet or a section of a spherical mirror or cylinder) has motivated a large amount of literature in science and engineering with applications to microwave measurement [1], computer vision [2], design tolerance [3], metrology and inspection of mechanical parts [4,5], archaeology [6], geodesy [7], and many others. For example, the manufacturing precision for a camshaft in an internal combustion engine significantly correlates to the performance of the engine [8].

In recent years, a significant issue has been identified with automated inspection techniques that utilize measurement equipment such as coordinate measuring machines (CMMs) [3]. For example, to determine the diameter of a cylinder, a CMM is programmed to acquire points on the surface and then optimize the parameters for best-fit geometry. This type of measurement can be affected by machine variation, part variation, inaccurate measurement algorithms, and the fitting algorithm employed by the machine control unit [9,10,11].

Typically, to improve geometric fitting, one samples a larger number of points to reduce the influence of outliers and minimize the effect of measurement errors. Depending on the application, there are many optimization criteria used to determine the parameters; these criteria vary with the method used to determine the parameters and with the statistical error model [12]. The fitting problem could be classified into algebraic and geometric methods [13,14]. The algebraic method uses the implicit form and minimizes the residuals, while the algebraic method minimizes the sum of squares for the distance between scattered points and the conic. Examples of geometrical methods include minimizing the mean square error (MSE) sum, the inversion method of Brandon and Cowley [15], the minimum-circumscribed (MC), and maximum-inscribed (MI) criteria [16]. The most widely used method, which minimizes the sum of squared deviations (i.e., least square (LS) fitting), is algebraic fitting. This method is not sensitive to outliers and it is relatively easy to implement. An example of a robust method is the Hough transform, which is mainly used in digital imaging [13], full LS, reduced LS, and modified LS [1], and Delogne-Kasa for noisy data [17]. When fitting a circle, the LS methods result in non-linear and implicit equations that can only be solved numerically, often following linearization techniques [4].

Since the above statistical fitting methods rely on the availability of excessive data points that exceed the number of unknown parameters in the substitute geometry model, these methods are not conducive to supporting the needs of mass production. Therefore, the intent of this research is to introduce an efficient, economical method to assess part geometry in mass production environment. Furthermore, there is a need for an accurate method that can utilize a reduced set of data points. The literature review reveals a lack of research in this area. For example, the National Institute of Standards and Technology (NIST) found that three-point measurement procedures depend only on the mean and variance of point measurement errors and are essentially independent of their statistical distribution [18].

Since the CMM has been the most common technique used for inspection of an arc circle on mechanical parts, the assumptions relative to circle fitting are currently based on data measured from one reference coordinate. Although a CMM provides an accurate reconstruction of complex shape from scattered points, conventional CMM measurement is cumbersome and often fails to meet the needs of mass production. This is primarily due to how a CMM carries out the measurement process. A CMM system carries out the measurement serially from one point to another by a sensor attached to CMM arm, which can lead to measurement errors. The purpose of this paper is to develop an efficient, automated method and apparatus for evaluating the radius of parts on a mass production line by measuring the three points from displacement sensors located within an independent coordinate system.

The paper is organized as follows: Section 2.1 derives a closed-form solution algorithm for calculating a circle and radius from the data points measured by three independent Cartesian coordinate systems. Section 2.2 introduces a calibration algorithm and procedure to recover the relative location of the coordinate system. Section 2.3 proposes the procedure to measure roundness of a circular part based on previous methods. Numerical simulation studies are conducted to validate the proposed method by using a mathematical model in Section 3.1, Section 3.2, and Section 3.5, and CAD simulation in Section 3.3, Section 3.4, and Section 3.6. Lastly, Section 3.7 describes a simplified experiment that validates the proposed methods.

## 2. Materials and Methods

Suppose one would want to inspect a unit circle by measuring three points spaced randomly over an arc as shown in Figure 1. The goal is to develop a fitting technique from the set of points measured by uncorrelated distributed displacement sensors. Thus to accomplish this, divide the fitting technique into two distinct methods:Three-point inverse kinematic algorithm (TPIK): a calibration procedure that references the local coordinates of each sensor into one global coordinate system (G). This produces system parameters whose values are based on a numerical solution of mixed non-linear equations. Further discussion to follow in Section 2.2.Three-point forward kinematic algorithm (TPFK): a closed form solution that algebraically fits the transformed data points through the circumference of a circle. Further discussion to follow in Section 2.1.

The TPFK algorithm uses the standard representation of circle with a radius and a center. The goal is to extract the radius regardless of where the center is located relative to the global coordinate system. The TPIK algorithm suggests deployment of a calibration master of known geometry to estimate location of the sensors. Additionally, TPIK provides a mean for an operator to calibrate the overall offset caused by the measurement system, which could be due to linear or non-linear shifts in sensor reading, or unknown relative movements of sensors. Therefore, it is recommended to carry out periodic calibration to avoid accumulation drifts in the measurements.

### 2.1. Three-Point Forward Kinematics (TPFK) Algorithm—Recovery of Circle Geometry

The TPFK algorithm is a deterministic method that calculates the radius R and the center C(a,b) of a circle from point coordinates measured by spatially distributed displacement sensors $\left\{s_{1},\;s_{2},\;s_{3}\right\}$ shown in Figure 1. The sensors are arranged to measure the vertical displacements $\left\{y_{1},\;y_{2},\;y_{3}\right\}$ relative to their local coordinates. The position of the two local coordinate sensors $\left\{s_{2},\;s_{3}\right\}$ are measured with respect to a global coordinate system {G}, which we chose to fix at sensor $\left\{s_{1}\right\}$. Therefore, the vector positions of the points located at the arc $\left\{P_{1},\;P_{2},\;P_{3}\right\}$ can be expressed in reference to the global coordinate system as $\left\{P_{1}\left(0,y_{1}\right),\;P_{2}\left(x_{2o},y_{2o}+y_{2}\right),\;P_{3}\left(x_{3o},y_{3o}+y_{3}\right)\right\}$. The radius is defined by the linear distance between any point on the circle and its center C(a,b), thus can be expresses as:(1)$${\left|P_{1}C\right|}^{2}={\left|P_{2}C\right|}^{2}={\left|P_{2}C\right|}^{2}=R^{2},$$ where the linear distance between a point on a circle P(x,y) and its center is simply obtained from three data points.
(2)$$R_{3p}=\left|\sqrt{{\left(a-x\right)}^{2}+{\left(b-y\right)}^{2}}\right|.$$

The circle center C(a,b) is referenced to the global coordinate. Therefore, rewriting the Equation (1) yields:(3)$${\left|P_{1}C\right|}^{2}={\left|P_{2}C\right|}^{2}\to {\left(a\right)}^{2}+{\left(b-y_{1}\right)}^{2}={\left(a-x_{2o}\right)}^{2}+{\left(b-y_{2o}-y_{2}\right)}^{2},$$
and
(4)$${\left|P_{1}C\right|}^{2}={\left|P_{3}C\right|}^{2}\to {\left(a\right)}^{2}+{\left(b-y_{1}\right)}^{2}={\left(a-x_{3o}\right)}^{2}+{\left(b-y_{3o}-y_{3}\right)}^{2}.$$

Upon expanding and simplifying the Equations (3) and (4), the results is a linear equation in matrix form $M\vec{C}=B$:(5)$$\underset{M}{\underbrace{\left[\begin{array}{cc} 2x_{2o} & \left(-2y_{1}+2y_{2}+2y_{2o}\right)\\ 2x_{3o} & \left(-2y_{1}+2y_{3}+2y_{3o}\right) \end{array}\right]}}\underset{C}{\underbrace{\left[\begin{array}{c} a\\ b \end{array}\right]}}=\underset{B}{\underbrace{\left[\begin{array}{c} x_{2o}^{2}-y_{1}^{2}+y_{2}^{2}+2y_{2}y_{2o}+y_{2o}^{2}\\ x_{3o}^{2}-y_{1}^{2}+y_{3}^{2}+2y_{3}y_{3o}+y_{3o}^{2} \end{array}\right]}}.$$

Upon solving for $\vec{C}\left(a,b\right)=M^{-1}B$, a point can be used in the circle such as $P_{1}\left(0,y_{1}\right)$ to calculate circle radius from Equation (2).

### 2.2. Three-Point Inverse Kinematics (TPIK) Algorithm—Calibration Procedures

The TPFK problem discussed in the previous section requires prior knowledge of the position $S_{p}\left\{x_{2o},y_{2o},\;x_{3o},y_{3o}\right\}$, which is the origin of the sensors $\left\{s_{2},\;s_{3}\right\}$. At the origin, one assumes that all initial displacement $\left\{y_{1},\;y_{2},\;y_{3}\right\}$ values are reset or forced to read zero. Often, for various reasons, it is impractical to obtain the S values by using direct measurement. Barriers to direct, efficient measurement include a lack of proper tooling, desired level of measurement accuracy is not obtainable, excessive set-up time, set up is not conducive for the production environment, measurement system idle time attributed to quality protocol, and change of operator and/or the part. As a result, there is a need to conduct a periodic calibration procedure that resets the errors because of the controlled or the uncontrolled factors that result in changing the value of S. These factors could be attributed to the mechanical movement of sensors, retooling, thermal expansion, changes in the characteristics of the sensors, and many other unpredictable factors. Therefore, in this section a systematic calibration procedure is proposed that inversely calculates $S_{p}\left\{x_{2o},y_{2o},\;x_{3o},y_{3o}\right\}$, which must remain unchanged during measurement. The $S_{p}$ represents part of the system parameters that are later plugged into Equation (5) to calculate the center of the circle $\vec{C}\left(a,b\right)$, and its radius by using Equation (2). It should be noted that TPFK as well as TPIK (will be shown later in this section) do not require information about the center of the calibration master.

A calibration master was proposed and depicted in Figure 2a. The calibration master is designed with a minimum of two different concentric radii. For example, The calibration master in Figure 2a is fabricated with three known concentric radii $\left\{R_{c1},R_{c2},\;R_{c3}\;\right\}$, whose design values depend on the size of the part being inspected and the range of the sensors used. The fabrication of a good calibration master with precise $\left\{R_{c2},\;R_{c3}\right\}$ is crucial to the success of the calibration and can be achieved either by additive manufacturing, such as 3D printing, or subtractive fabrication such as machining the calibration master from a thick cylinder shown in Figure 2b. However, the fabrication cannot guarantee accurate design values, therefore it is necessary to measure the radii $\left\{R_{c2},\;R_{c3}\right\}$ after fabrication at given operating conditions. The value of $R_{c1}$ is not part of the calculation—as we will see later—however, it is used in the calibration to initialize the location of the sensors, S. The center of the calibration master $C^{c}$ relative to any global coordinate system *G* (G was chosen to be placed at sensor s_1_ as shown in Figure 1) should remain unchanged during the calibration procedure.

To better understand the design requirement of the calibration master in relation to the inspected part, it was assumed that the minimum sensing range was $\Delta L_{min}$. Thus, the nominal radius value of the inspected part was ~$R_{a}$, and the calibration master should be $\left(R_{c3}-R_{c1}\right)<\Delta L_{min}$ and $\;R_{c3}>R_{a}>R_{c1}$. The sensors were arranged in parallel such that each arc in the calibration master could be measured at once by all three sensors as shown in Figure 2c. The sensors were made adjustable in the *y*-direction to allow inspection of a wider range of parts. The shared center $C^{c}$ of the calibration master must be tightly secured in a fixed rotating axis; allowing the measurement of displacement for every arc segment while their shared center remains unchanged.

Following the discussion above, the derivation of the TPIK algorithm can be shown by using the following example: Assume that spring-loaded probes in Figure 2c are brought on contact and then pushed against the $R_{c1}$ segment of the calibration master. This is to maximize the reading range of the sensors. Each sensor is fastened onto the fixture and then reset to read zero displacement. This step has established $S_{p}\left\{x_{2o},y_{2o},\;x_{3o},y_{3o}\right\}$, however the values are yet to be found. Equation (5) is applied to evaluate the center of the calibration master $C^{c}$ by either using segment $R_{c2}$ or $R_{c3}$. Substituting the known $R_{c2}$ value yields the following matrix equation:(6)$$\underset{C_{c2}}{\underbrace{\left[\begin{array}{c} a_{c}\\ b_{c} \end{array}\right]}}=\underset{A_{c2}}{\underbrace{{\left[\begin{array}{cc} 2x_{2o} & \left(-2y_{12}+2y_{22}+2y_{2o}\right)\\ 2x_{3o} & \left(-2y_{12}+2y_{32}+2y_{3o}\right) \end{array}\right]}^{-1}}}\underset{B_{c2}}{\underbrace{\left[\begin{array}{c} x_{2o}^{2}-y_{12}^{2}+y_{22}^{2}+2y_{22}y_{2o}+y_{2o}^{2}\\ x_{3o}^{2}-y_{12}^{2}+y_{32}^{2}+2y_{32}y_{3o}+y_{3o}^{2} \end{array}\right]}}$$ where $\left\{y_{12},y_{22},y_{32}\right\}$ displacement are measured on the $R_{c2}$ segment. The above equation breaks down into its element form:(7)$$C_{c2}\left(1\right)-\left[A_{c2}B_{c2}\right]\left(1\right)=0,\;\mathrm{and}\;C_{c2}\left(2\right)-\left[A_{c2}B_{c2}\right]\left(2\right)=0.$$

In addition, substitute the center and radius of the $R_{c2}$ segment in Equation (2) to yield:(8)$$R_{c2}=\underset{K_{c2}}{\underbrace{\left|\sqrt{{\left(a_{c}\right)}^{2}+{\left(b_{c}-y_{12}\right)}^{2}}\right|}},$$ and write as:(9)$$R_{c2}-K_{c2}=0.$$

Similarly, for $R_{c3}$, we can write:(10)$$\underset{C_{c3}}{\underbrace{\left[\begin{array}{c} a_{c}\\ b_{c} \end{array}\right]}}=\underset{A_{c3}}{\underbrace{{\left[\begin{array}{cc} 2x_{2o} & \left(-2y_{13}+2y_{23}+2y_{2o}\right)\\ 2x_{3o} & \left(-2y_{13}+2y_{33}+2y_{3o}\right) \end{array}\right]}^{-1}}}\underset{B_{c3}}{\underbrace{\left[\begin{array}{c} x_{2o}^{2}-y_{13}^{2}+y_{23}^{2}+2y_{23}y_{2o}+y_{2o}^{2}\\ x_{3o}^{2}-y_{13}^{2}+y_{33}^{2}+2y_{33}y_{3o}+y_{3o}^{2} \end{array}\right]}},$$ where $\left\{y_{13},y_{23},y_{33}\right\}$ displacement is measured on the $R_{c3}$ segment. The above equation breaks down into its element form:(11)$$C_{c3}\left(1\right)-\left[A_{c3}B_{c3}\right]\left(1\right)=0,\;\mathrm{and}\;C_{c3}\left(2\right)-\left[A_{c3}B_{c3}\right]\left(2\right)=0.$$

In addition, substitute the center and radius of the $R_{c3}$ segment in Equation (2) to yield:(12)$$R_{c3}=\underset{K_{c3}}{\underbrace{\left|\sqrt{{\left(a_{c}\right)}^{2}+{\left(b_{c}-y_{13}\right)}^{2}}\right|}},$$ which is expressed as:(13)$$R_{c3}-K_{c3}=0.$$

It must be noted that $C_{c2}\left(1\right)=$
$C_{c3}\left(1\right)=a_{c}$, and $C_{c2}\left(2\right)=C_{c3}\left(2\right)=b_{c}$. Therefore Equations (7), (9), (11), and (13) provide six non-linear equations with a total of six unknowns $S\left\{x_{2o},y_{2o},\;x_{3o},y_{3o,},\;a_{c},b_{c}\right\}$. A successful solution will provide an indirect way for calculating system parameter $S=\left\{x_{2o},y_{2o},\;x_{3o},y_{3o,}\right\}$, which is sufficient for the TPFK algorithm. However, the subset solution $\left\{a_{c},b_{c}\right\}$ is unnecessary for the TPFK algorithm, and is only needed to complete the computation of TPIK algorithm. Likewise, the center of the inspected part in the TPFK algorithm is implicit in the computation of the radius where it can be introduced to the measurement system from any location as long as the sensors can read any three points on the surface. Consequently, fitting a circle by using these proposed methods reduces handling errors and reduces the measurement time.

Finally, it should be noted that there is no restriction on how the sensors are initialized, i.e., it is possible to replace the circular shape $R_{c1}$ in the calibration master with a flat surface to level the offsets between the sensors {$\left|y_{1o}-y_{3o}\right|$
$\approx \left|y_{1o}-y_{2o}\right|\approx 0$}. This could help the numerical solution of the TPIK algorithm to converge faster to a solution as will be observed later in the simulation sections.

### 2.3. Roundness Index Based on TPIK and TPFK Algorithms

Roundness was defined in the index in this paper by the average nominal radius $R_{a}$ that minimizes the square summation of residual errors for every arc-segment in Figure 3a whose radius $R_{3p,i}$ is computed by using TPIK and TPFK algorithms. This measure is given by:(14)$$W=\sum_{i=1}^{N}{\left(R_{a}-R_{3p,i}\right)}^{2},$$ where *N* is the total number of arc-segments. The optimization of Equation (14) with respect to $R_{a}$, simply gives the average $R_{a}=\overset{N}{\underset{i=1}{\sum }}R_{3p,i}/N$ for the best fitted circle in Figure 3b. Similarly, the average center point $C_{a}\left(a,b\right)$ is $C_{a}\left(\overset{N}{\underset{i=1}{\sum }}a_{3p,i}/N,\overset{N}{\underset{i=1}{\sum }}b_{3p,i}/N\right)$, where $a_{3p,i}$ and $b_{3p,i}$ are obtained for each arc-segment in the TPFK algorithm. Roundness index is defined by $\beta =1-\sqrt{W/N}/R_{a}$, where a perfect unit circle has a value of 1. Alternatively, one can define $\beta^{*}=1-\sqrt{W/N}/R_{e}$ when the exact nominal radius $R_{e}$ is known. It should be noted that in ideal scenarios, $R_{a}=R_{e}$ when all arc profiles has no perturbation, as will be shown in later sections. The computation of a β or $\beta^{*}$ requires a onetime calibration via the TPIK algorithm, and then the application of TPFK algorithm to every arc-segment around the particular part being inspected. The characteristics of the roundness as function of perturbation pattern(s) will be presented in later section. One advantage of this definition is that the roundness is well suited for the automatic inspection because it does not require a common center C(a,b) for all arc-segments, and can be achieved by various sampling patterns. Examples of sampling patterns encompasses non-overlapping or overlapping arc-segmentations as shown in the three cases in Figure 4. Numerical studies will be conducted in later sections to show the effect of the sampling angle θ on the roundness index.

## 3. Results and Discussion

### 3.1. Validation of the TPIK Algorithm by Mathematical Simulation

In this section, a mathematical model was constructed to simulate the *y*-displacements between sensor coordinates and parametric circles. For simplicity, it was assumed that the sensors were initialized at the same level, i.e., $y_{1o}=y_{2o}=y_{3o}=0$. The calibration master was represented with parametric equations (x(t),y(t)) that simulate two concentric circles of a known center and two given radii $\left\{R_{c2},\;R_{c3}\right\}$ as shown in Figure 5. To find the *y*-displacements between the sensor coordinates and the parametric equations, substitute the *x*-location in $t=\cos^{-1}\left(\left(a_{c}-x\right)/R\right)$ to find *t*, and then compute displacements from $y=b_{c}-R\sin\left(t\right)$. Typically, for a circle, there are two solutions obtained for *t*, and one must select the solution, which corresponds to the smallest y. The *y*-displacement results are summarized in Table 1.

In practice, it is desired to recover the solution values $S\left\{x_{2o},\;x_{3o},y_{2o},y_{3o},\;a_{c},b_{c}\right\}$ by using the displacement information $D\left\{y_{12},y_{22},y_{32},y_{13},y_{23},y_{33}\right\}$ in the TPIK Equations (7), (9), (11), and (13). It is important to reemphasize here that the solution set $S_{p}$ is sufficient for TPFK algorithm, but the S solution set is necessary to compute $S_{p}$. The TPIK algorithm is comprised of non-linear equations, which were intentionally solved iteratively using optimization techniques. The ‘fsolve’ command in MATLAB [19] was utilized along with the ‘trust-region-dogleg’ algorithm and the ‘optimplotfirstorderopt’ option to plot first-order optimality at each iteration. This numerical method requires an initial estimation for a solution, *S*. The validity of the solution depends on the convergence criterion of the selected optimization method and the final values of the functions in Equations (7), (9), (11), and (13). The numerical iterations of the optimization method might prematurely stop producing multiple final values depending on the convergence criterion and initial estimation. Therefore, the convergence of the solution was studied and several initial estimations benchmarked relative to true solution that could be obtained from the parametric model. Table 2 is a summary of the studies conducted for several initial estimations. In the absence of information about the true solution, success was evaluated form the numerical solution using its convergence to a final solution and the residual that is defined from Equations (7), (9), (11), and (13) by:(15)$$E_{rs}={\left(C_{c2}\left(1\right)-\left[A_{c2}B_{c2}\right]\left(1\right)\right)}^{2}+{\left(C_{c2}\left(2\right)-\left[A_{c2}B_{c2}\right]\left(2\right)\right)}^{2}+\;{\left(C_{c3}\left(1\right)-\left[A_{c3}B_{c3}\right]\left(1\right)\right)}^{2}+{\left(C_{c3}\left(2\right)-\left[A_{c3}B_{c3}\right]\left(2\right)\right)}^{2}+{\left(R_{c2}-K_{c2}\right)}^{2}+{\left(R_{c3}-K_{c3}\right)}^{2}.$$

The *E_rs_* value could be used as an indicator to evaluate the difference between the numerical solution and the true solution if the solution exists and is unique. While the existence and uniqueness of a real solution for both the TPIK algorithm and the optimization method was not investigated, the researcher purposely limited this paper to the numerical comparison. The numerical values of S in Table 2 are rounded to 3 decimal points and displayed for each trial when the optimization algorithm converges or stops. For example, trials 1, 4, 5, 8, and 9 in Table 2 produced acceptable results because they converged quickly with very small *E_rs_* values. While trials 6, 7, and 10 failed to produce acceptable results due to their non-convergence and very large *E_rs_* values. On the other hand, trials 2 and 3 did not converge, however they produced adequate final values because *E_rs_* were considerably small when compared to trials 6, 7, and 10. These studies also show that the numerical solution in trial 8, which has the fastest convergence and smallest *E_rs_*, had the least difference from the true solution.

### 3.2. Validation of the TPFK Algorithm by Mathematical Simulation

As previously discussed, the TPFK algorithm requires that system parameters $S_{p}\left\{x_{2o},\;x_{3o},\;y_{2o},\;y_{3o}\right\}$ are previously known or computed from the TPIK algorithm. It is assumed that the displacement measurements of the inspected part are the same as the values in Table 2. The TPFK algorithm performance was compared by using two sets of system parameters: true solution and the computed solution. Table 3 shows that the computed {*a*, *b*, *R*_3*p*_} values for both radii agree closely with the true radii values.

### 3.3. Validation of the TPIK Algorithm by CAD Simulation

A 3D assembly CAD model was constructed using SolidWorks software [20] and a dimensioning accuracy of ±0.01 mm. The model consists of three identical sensor probes with round tips, a fixture, a disc-part of unknown radius (inspected part), and a calibration master of three known radii as shown in Figure 6. The mating constraints are applied such that the probes can slide, and the calibration master can freely rotate. The tip of each probe is brought to contact at the largest radius ($R_{c1}=\;$40 mm) in the calibration master. The probes location is fixed, and the calibration master is rotated about its center to measure displacements corresponding to each radius in the calibration master. The tip of sensor 1 is used as a reference in the Cartesian coordinate system to measure the displacements $\left\{y_{12},y_{22},y_{32}\right\}$ and {$y_{13},y_{23},y_{33}\}$ for $\;R_{c2}$ and $\;R_{c3}$, respectively. The measurements are recorded in Table 4. The TPIK algorithm was used to compute $\left\{\left(a_{c},b_{c}\right),\left(x_{2o},y_{2o}\right),\left(x_{3o},y_{3o}\right)\right\}$ for several initial estimations and the measurement resolution of the displacements results are shown in Table 5. A comparison of these values with the true dimensions extracted from the 2D-CAD drawing of the assembly is shown in Figure 7. In this simulation, the measurement resolution of the displacement, *n*, was achieved by rounding dimensions in Table 4.

It is observed the convergence to a solution in trial 3 of Table 5 was unsuccessful despite the high measurement resolution used in the computation. This could be attributed to using initial estimations that are far-off from the true values or because the optimization algorithm might not perform adequately when it searches for non-zero values. Therefore, to improve the convergence, it is recommended to initialize the *y*-locations of the sensors 2 and 3 $\left\{y_{2o},y_{3o}\right\}$ to zero by replacing the large radius in the calibration master with a flat surface. However, trials 1 and 2 did not converge, yet produced an accurate solution when compared to the true values. This might be due to a relatively small *E_rs_* value. The effect of the measurement resolution was studied using the initial estimations to set the true values. This was carried out for *n* = {6,5,4,2} in trials 4–8, which confirms that high measurement resolution was not always necessary to attain an acceptable solution. Similarly, a low-resolution measurement may successfully converge to a solution, but is far off from the true values. For example, trial 5 was rounded to *n* = 5 decimal places, and the optimization had successfully converged to the true values with a small *E_rs_*. The simulation was repeated using the same initial estimation with *n* = {4,3,2}. Interestingly, trial 7 was simulated for *n* = 3 (trial 5) and had not only converged quicker than that of *n =* 5 (trial 3), but also had a smaller *E_rs_* value. However, the solution for trial 7 was farther off from the true value when evaluated against trial 5.

### 3.4. Validation of the TPFK Algorithm by CAD Simulation

It was presented in previous sections that the TPFK algorithm computed the radius of the part independent of how it was positioned. In this study, the true location of the sensors was obtained {$x_{2o}=10.000,\;x_{3o}=20.000,\;y_{2o}=-1.175,\;y_{3o}=-0.014$} mm and the true inspected part geometry by using CAD dimensioning tools in Figure 6 where all values were approximated to *n = 3* decimal points. The first group of simulations was conducted for inspected parts whose true radii *R_e_* ranged between 38.000 mm to 39.500 mm and with all of their true centers fixed at (10.010, 38.788) mm. The TPFK algorithm was implemented to compute the “fit” of their radii and centers based on the displacement measurement and the sensor location.

The absolute error between the true and computed radius $\left|R-R_{e}\right|$ was calculated and the results are summarized in Table 6. The error increased as the radius of the inspected part radius approached 40 mm. Specifically, when the *y*-displacements measurements approached zero, the TPFK became sensitive to rounding. This could be attributed to sensors that were initialized for a calibration master whose radius was at 40 mm.

The second group of study was conducted for an inspected part of a fixed radius of 38 mm, however, the inspected part was introduced to the sensors from several arbitrary locations. It was determined that the computed radius of the inspected part was in close proximity with the true value for the arbitrary locations. Additionally, the $\left|R-R_{e}\right|$ error followed the same findings as in the first study group where the error generally increased when the displacement measurement vector decreased.

### 3.5. Study of the Roundness Index Based on the Mathematical Model

The fundamental aim of this section was to determine how well the proposed TPIK and TPFK algorithms work in computing the roundness index β of the inspected part. It was assumed that the inspected part had perturbation around the circler profile. Additionally, it was desired to observe the effect of the sampling pattern on the roundness index. Furthermore, it was assumed that the inspected part is fixed at the center $C_{e}\left(a,b\right)$ and was able to rotate in increments of δθ with a total rotational angle θ measured from a given reference. The perturbed circle was modeled by using the nominal radius $R_{e}$ and a sinusoidal function as shown in Figure 8. The actual radius value of a point at an angle of φ can be represented by a piecewise smooth curve in reference to its polar coordinate while it is not in rotation (θ=0):(16)$$R\left(\varphi \right)=R_{e}+R_{p}\times \sin\left[\varphi \times \omega \left(\theta \right)\;\right],$$ where $R_{p}$ is the amplitude of the fluctuation of a sinusoidal function, and ω(θ) is the frequency function. The first step is to estimate the number of sample points required for scanning a fluctuating profile. Theoretically, there are infinite points along the length of the inspected part profile. The total arc length L of R(φ) is computed over the entire interval 2π:(17)$$L=\int_{0}^{2\pi }\sqrt{{\left(R\left(\varphi \right)\right)}^{2}+{\left(\frac{\partial R\left(\varphi \right)}{\partial \varphi }\right)}^{2}}d\varphi .$$

Let the desired linear increment ΔL between any two data points on the surface of the inspected part be used to sample the entire length *L*. This generates approximately ~L/ΔL sample points over the entire length. Suppose a circle (is utilized) whose circumference is equal to the length *L*, or has a radius of $R_{L}=L/2\pi$, then the average angular increment that uniformly dissects this circle is $\delta \theta^{*}≅\Delta L/R_{L}$ radian, or $\delta \theta^{*}≅2\pi \Delta L/L$ radian. The significance of $\delta \theta^{*}$ is to provide the programmer a reference value for selecting δθ. Likewise, the reference value can be used to calculate the spacing between two points when a uniform rotational increment sampling is imposed.

The function is parameterized to simplify the analysis in Equation (16) as follows:(18)$$x\left(t\right)=\left(R_{e}+R_{p}\sin\left[t\omega \left(t\right)\right]\right)\cos\left(t+\theta \right)-a\;\;\;\;\;\;\;\;2\pi \geq t\geq 0,\;\mathrm{and}\;y\left(t\right)=\left(R_{e}+R_{p}\sin\left[t\omega \left(t\right)\right]\right)\sin\left(t+\theta \right)-b\;\;\;\;\;\;\;\;\;2\pi \geq t\geq 0,$$ where θ is the rotational angle $360^{o}$
≥θ≥0 with δθ increment. Figure 9a simulates several fluctuating circles by using a non-zero frequency and fluctuation amplitude. The sensor displacements $y_{1},\;y_{2},\;\mathrm{and}\;y_{3}$ shown in Figure 9b represent the intersection between the vertical line passing between each sensor and the inspected part. It is desired to observe how the roundness index varies with the frequency, fluctuation amplitude $R_{p}$, nominal radius $R_{e}$, and measurement resolution *n*. Table 7 shows an example of the simulation carried out for $\delta \theta =20^{o}$, frequency ω=5 rad, $R_{p}$= 38/40 mm, $R_{e}\;$= 38 mm, *a* = 10 mm, *b* = 40 mm, and *n* = 3. The sensor locations shown in Figure 9b were set to $x_{1o}=0,{\;x}_{2o}=10.000\;\mathrm{mm},{\;x}_{3o}=20.000\;\mathrm{mm},{\;y}_{1o}=0,{\;y}_{2o}=0\;\mathrm{mm},{\;\mathrm{and}\;y}_{3o}=0$. The simulation was run as follows: The initial step was to solve Equation (18) at every angle θ to find the solution set *t* for which $x\left(t_{1}\right)=0,\;x\left(t_{2}\right)=x_{2o}=10\;\mathrm{mm},$ and $\left(t_{3}\right)=x_{3o}=20\;\mathrm{mm}$. The solution requires a numerical search algorithm such that $y\left(t\right)<b\;\forall t\in \left\{t_{1},t_{2},t_{3}\right\}$. Thus, the solution lies on the lower half of the part shown in Figure 9b where $\left\{y\left(t_{1}\right),y\left(t_{2}\right),y\left(t_{3}\right)\right\}$ represent the displacement distances between sensors and the part surface, i.e., $\left\{y_{1},y_{2},y_{3}\right\}$. The second step was to apply the TPFK algorithm to calculate the corresponding part radius and its center from these three displacement vectors. The previous steps were repeated for every θ; the results are shown in Table 7. The average center of the inspected part was $\left(\bar{a},\bar{b}\right)=\left(9.9994,\;39.9821\right)$ mm, and the equivalent average radius was $R_{a}$= 37.9992 mm. These values were obtained for the inspected part whose center was (10,40) mm, nominal radius $R_{e}=38\;\mathrm{mm}$, and fluctuation amplitude $R_{p}$ = 38/40 mm. Therefore, stochastically, the average radius $R_{a}$ could be used for describing the radius of the inspected part, and the fluctuation of $R_{3p,i}$ values could be used as an indication of roundness. Comprehensive studies of the relationship between roundness β and the fluctuation ratio $R_{e}/R_{p}$ were simulated for several conditions in Figure 10. For example, Figure 10a shows a simulation of the roundness index for three sampling increments δθ ={0.5°,10°,60°} and ω=50 rad. It is evident that the computation of the roundness index was sensitive to the sampling increment when the fluctuation amplitude was large, but when the fluctuation was small, all roundness indices converged to the same final value regardless of the value of δθ.

This implies that in such cases, fewer sampling points could be used to calculate the roundness of the inspected part. Under the same conditions as applied to Figure 10a, the effect of the fluctuation ratio $R_{e}/R_{p}$ to the ratio between average radius to the exact radius $R_{a}/R_{e}$ was observed with results shown in Figure 10b. It was determined that the $R_{a}$ converged to $R_{e}$ as the fluctuation amplitude became small, however, this convergence was rapid when the frequency of the fluctuation decreased as noted by comparing Figure 10b,d.

In addition, a comparison between Figure 10a,c show that the computation of the roundness index was less sensitive to δθ when ω was small. Finally, the relation between the roundness index and the measurement resolution of the displacement was studied for *n* = {6,5,4,3,2} and were all found to be identical. This indicates that it could be possible to evaluate the roundness of the inspected part using a less expensive sensor with lower resolution.

Since the fluctuation had a symmetric pattern, the computation of average center {$\bar{a}=9.9994$, $\bar{b}=39.9821$} mm was close to the exact value {a=10, b=40} mm. Nevertheless, this did not imply that the difference between the average and exact values was predestined to worsen when the rotation increment increased. However, such a difference would also depend on the effect of the scanning resolution and the sample point distribution as will be presented in the subsequent section.

### 3.6. Study of Roundness Index Based on CAD Model

The “equation driven curve” function in SolidWorks was used to plot the parametric Equation (18). Figure 11a shows a solid model assembly comprised of three probes and a fluctuating part. To simplify the measurement in CAD software, the researcher simulated the fluctuation profile using Equation (18) in 2D drawing. Figure 11b shows an example of inspected part with $R_{e}/R_{p}=100$, and ω=20 rad/s. The center of the inspected part rotated freely. The displacements {$y_{1},y_{2},y_{3}\}$ were measured at every Δθ = 2^o^ by using the dimensioning tools in 2D drawing. Subsequently, the *{a, b,*
$R_{3p}$*}* were measured at every increment using the TPFK algorithm with results in Table 8.

Note that for increment Δθ = 2°, the measurements values repeated after every 16°, therefore the average values of {$\bar{a}$, $\bar{b}$, $R_{a}$} within the first J = 9 rotations would produce similar values if calculated from sample size M × J rotations, where M was any real integer. Therefore, one could reduce the cost of data collection when the surface pattern of the inspected part is known to have a uniform repetition. The model in Figure 11 produced an average radius $R_{a}\;$= 45.1944 mm and an average center of ($\bar{a},\bar{b})=$ (7.7273, 46.1588) mm. Although the sampling increment was small Δθ = 2, the average values differed from the exact value (a,b)=(7.873513, 43.48817) mm, and were also biased. Unlike the case study in Table 7 (ω=5 rad/s), which produced smaller differences despite the large sampling increment used (Δθ = 20). This could suggest that a smaller sampling increment be used when the fluctuation frequency is high.

### 3.7. Validation of the TPIK and TPFK Algorithms Based on the Empirical Study

A radial load bearing was mounted on a customized 3D printed fixture as shown in Figure 12 Three micrometers {S_1_, S_2_, S_3_} were used to measure the displacements from the initialized. coordinates into the surface of the bearing. Researchers applied adhesive tape of known thickness on the bearing surface to create multiple radii for both calibration and inspected part experiments.

The calibration procedures require two known radii sharing the same center, therefore, the base radius of the bearing $R_{c2}=40.00\;\mathrm{mm}$ was utilized to measure the first group of displacements, and also 2.13 mm thick tape was applied on another section of the bearing surface to create the second radius $R_{c3}=42.13\;\mathrm{mm}$ and obtain the second group of displacements. The average $\bar{y}$ and standard deviation *s_d_* measurements are summarized in Table 9. Likewise, following the calibration procedures an inspected part with radius of $R_{e}=41.03\;\mathrm{mm}$ was created by adding a 1.03 mm thick tape on a section on the bearing. Following this, the corresponding displacements were measured and results are shown in Table 10. A summary of all measurements is shown in the error plot in Figure 13.

The researcher used the displacement values of the calibration master in Table 9 to recover the system parameters $S\left\{x_{2o},{\;x}_{3o},{\;y}_{2o},{\;y}_{3o},a_{c},b_{c},\right\}$ by using the TPIK algorithm. While the measurement of *S* was difficult to obtain and inaccurate in practice, it roughly measured the sensors location and the center of the master calibration part $\tilde{S}${26,53,1,1,26,45} mm, and then these values were applied as the initial estimation in the algorithm to calculate S based on data in Table 9. The final values obtained from the TPIK algorithm after 106 iterations was S{25.30, 53.32, 1.12, 1.04, 25.71, 45.02} mm with a total numerical error *E_rs_* = 0.0071 mm. The final values were applied in the TPFK algorithm to calculate the radius of the inspected part using the data in Table 10. The estimated {*a*, *b*, *R*_3*p*_} values were {25.72, 44.75, 40.78} mm, respectively. This indicates that the error in measuring the inspected part radius was $er=\left|R_{3p}-R_{e}\right|=0.25$ mm or 0.61% percentage error, which represents acceptable agreement between values given the simplicity of the experiment setup used in this study.

## 4. Conclusions and Future Work

This paper developed practical methods to measure radius and roundness of a circular arc from three distributed gauges. The new proposed TPIK algorithm provides a calibration mechanism that resets the measurement error resulting from machine and human errors by computing the location of distributed gauges in reference to a global Cartesian coordinate. The TPIK algorithm is a non-stochastic fitting method but requires numerical optimization techniques to determine a solution. The solution was investigated based on iterative optimization techniques that depend on an initial estimation. While this paper was not concerned with the approach used in finding a solution, however, it was determined that the final value solution was sensitive to the initial estimation. The existence and uniqueness of solution have not yet been investigated mathematically, but it was verified geometrically, that there must be at least one solution to satisfy the six non-linear TPIK equations. In addition, a TPFK algorithm was offered, which is a closed form solution to compute the geometry of the circular arc from three distributed gauges. This approach is a modification of the direct algebraic method that uses three points to fit a circle, however, the algorithm itself contains estimated parameters from the TPIK algorithm. Due to simplicity, accuracy, and computational efficiency of the TPFK algorithm, it was not possible to compute the radius and the center of a part being inspected, but the method was to compute the roundness of a fluctuating surface. The proposed roundness index was based on aforementioned algorithms and demonstrated the capability to present the degree of circularity of a part. It was determined that a successful implementation of roundness index depends on the selection of a proper scanning rate for the waviness attributes of the part being inspected, which includes fluctuation amplitude and frequency. The degree of accuracy of measurement did not play a major role when the roundness was computed, however, the sample point distributions of the fluctuation had an additional impact. Future work will investigate numerical methods for solving the TPIK algorithms and the corresponding performance tradeoffs.

## Figures and Tables

**Figure 1 sensors-19-04679-f001:**
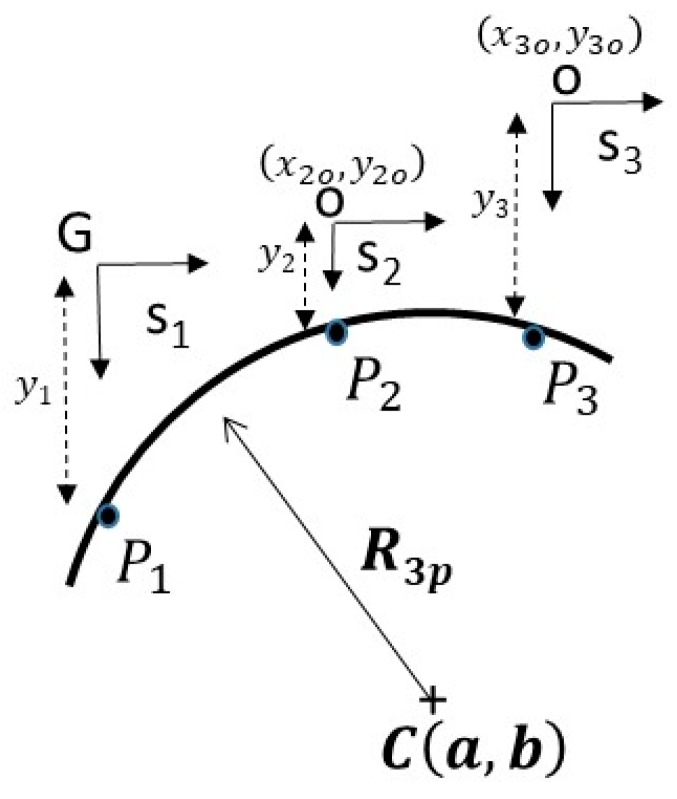
Three-point method: Measurement of the circle geometry by using three arbitrary points on an arc $\left\{P_{1},\;P_{2},\;P_{3}\right\}$ using global reference G located at $s_{1}.$ The sensors are arranged in parallel to reduce system complexity and set-up error.

**Figure 2 sensors-19-04679-f002:**
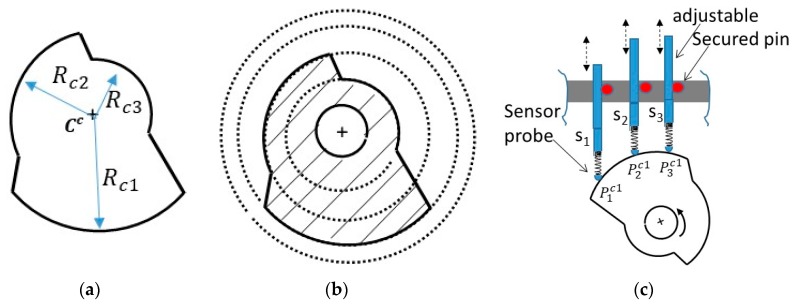
Calibration master: (**a**) consists of three concentric arcs $\left\{\;R_{c1}>R_{c2}>R_{c3}\;\right\}$ and one shared center $C^{c}$. (**b**) Manufacturability can be achieved by machining a thick cylinder. (**c**) A radial bearing insert will allow free rotation about a fixed center. Sensors are arranged vertically to allow a measurement of one radius at a time.

**Figure 3 sensors-19-04679-f003:**
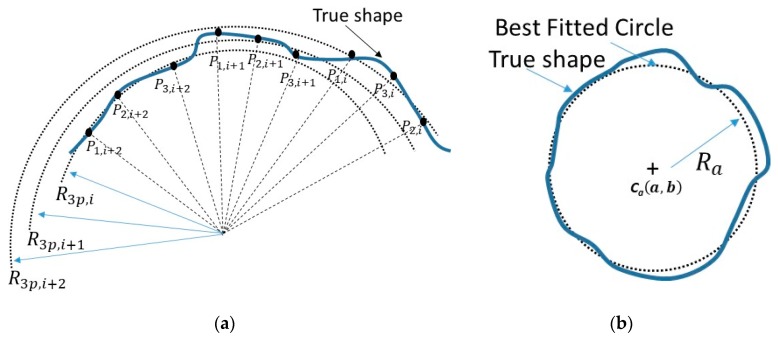
Roundness: measurement of roundness by using *N* number arc-segments. (**a**) Every arc-segment *I* is measured from three points {$P_{1,i},P_{2,i},P_{3,i}\}\;$ and has radius $R_{3p,i}$ obtained by the three-point forward kinematic algorithm (TPFK) and three-point inverse kinematic algorithm (TPIK) algorithms. (**b**) The best-fit radius and center of the inspected part by using all the segmentations.

**Figure 4 sensors-19-04679-f004:**
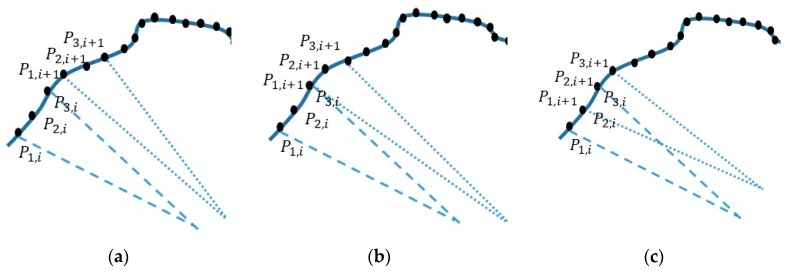
Sampling patterns: measurement of roundness with total of *M* number of sampled points. (**a**) Non-overlapping arc-segmentations, i.e., *M* = *3N*. (**b**) Overlapping arc-segmentations with one common point, i.e., *M* = *2N* + *1*. (**c**) Overlapping arc-segmentations with two common points, i.e., *M = N − 2*. Any two segments should not share same three points to avoid redundancy.

**Figure 5 sensors-19-04679-f005:**
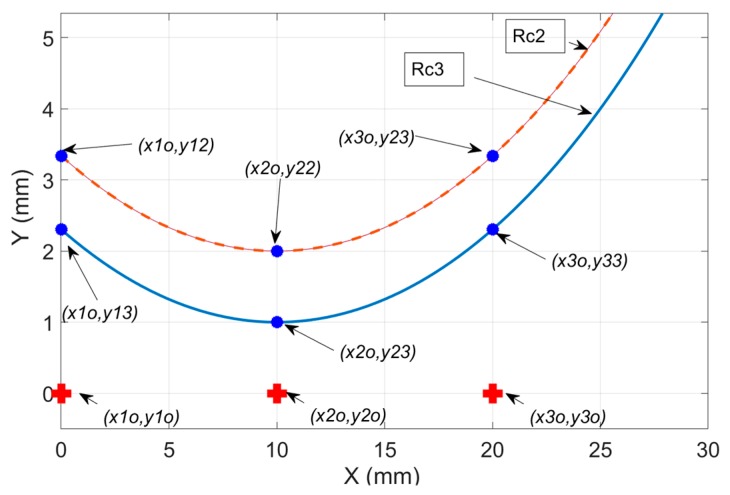
Mathematical simulation of calibration procedures: the circles are constructed for $\left(a_{c},b_{c}\right)=\left(10,40\right)\;\mathrm{mm}$ and two radii R={38, 36} mm by using two parametric functions $t=\cos^{-1}\left(\left(a_{c}-x\right)/R\right)$ and $y=b_{c}-R\sin\left(t\right)$.

**Figure 6 sensors-19-04679-f006:**
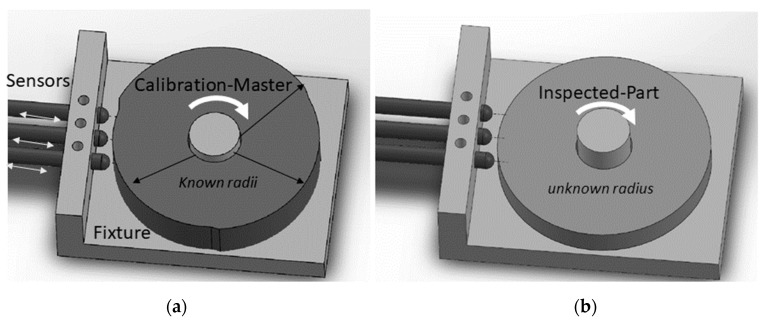
Validation of the TPFK/TPIK algorithms by CAD simulation: (**a**) calibration master assembly. (**b**) Inspected part assembly.

**Figure 7 sensors-19-04679-f007:**
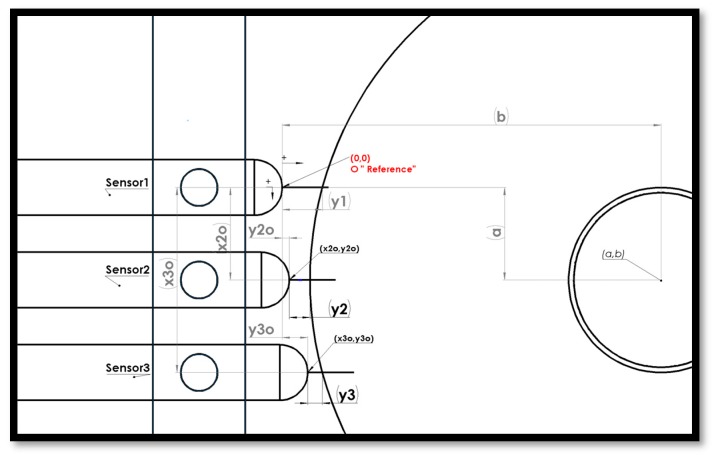
2D-CAD drawing of the assembly with annotations. The global reference is fixed at the tip of sensor 1 with y+ direction to the right and x+ direction downward.

**Figure 8 sensors-19-04679-f008:**
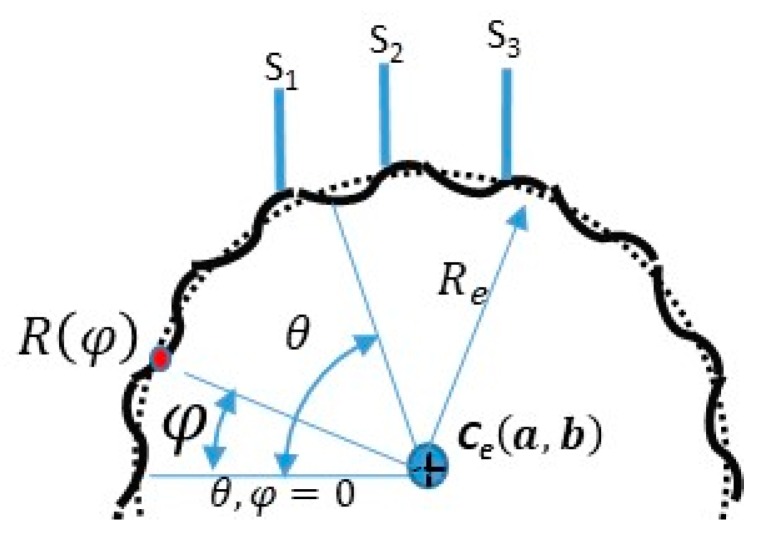
Perturbation of the inspected part.

**Figure 9 sensors-19-04679-f009:**
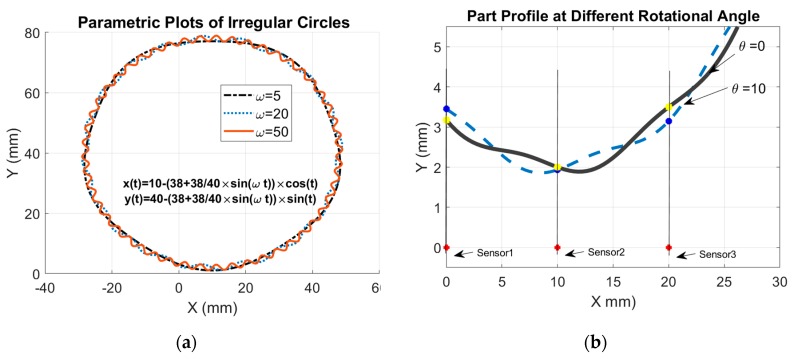
Mathematical representation of the inspected part with perturbation: (**a**) perturbed parts and (**b**) rotation of the inspected part at two angle, *θ* = 0° and *θ* = 10°.

**Figure 10 sensors-19-04679-f010:**
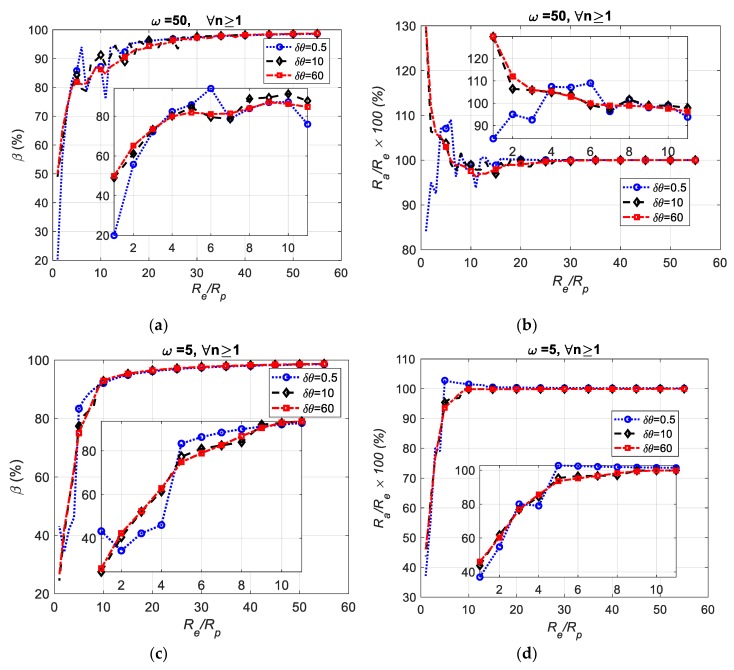
Exponential responses of the roundness index β simulated for the sampling increment θ, ratio of nominal radius to fluctuation amplitude $R_{e}/R_{p}$, and fluctuation frequency ω. (**a**) Roundness index vs. fluctuation ratio at frequency 50 rad/s. (**b**) Exact radius to nominal radius ratio vs. fluctuation ratio at frequency 50 rad/s. (**c**) Roundness index vs. fluctuation amplitude at frequency 5 rad/s. (**d**) Exact radius to nominal radius ratio vs. fluctuation amplitude at frequency 5 rad/s.

**Figure 11 sensors-19-04679-f011:**
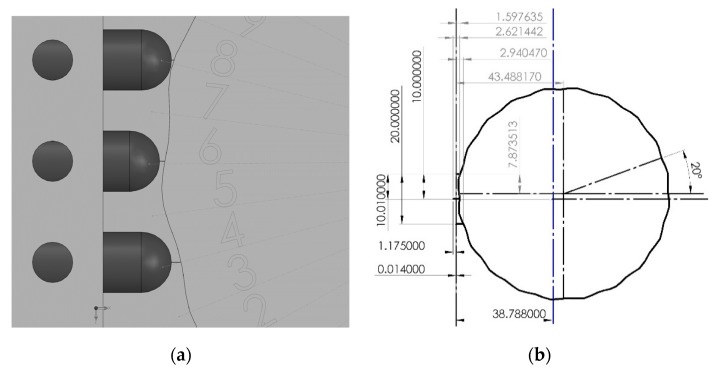
Roundness index based on the CAD model with $R_{e}=42.500$ mm, $R_{p}=0.425\;\mathrm{mm}$, $\omega \left(t\right)=20\frac{\mathrm{rad}}{s},\;a=7.873513\;\mathrm{mm},\;\mathrm{and}\;b=43.48817$ mm. (**a**) 3D-CAD model. (**b**) 2D-CAD drawing.

**Figure 12 sensors-19-04679-f012:**
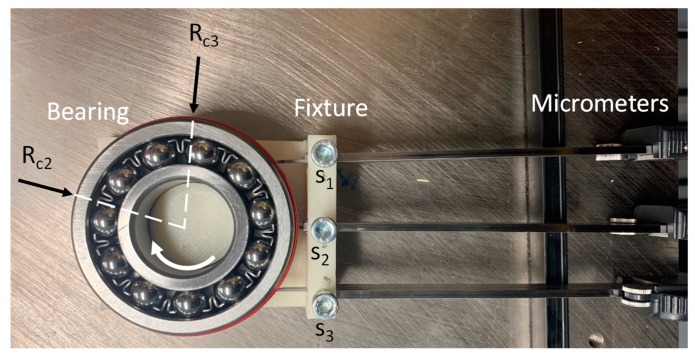
Apparatus for system calibration and characterization.

**Figure 13 sensors-19-04679-f013:**
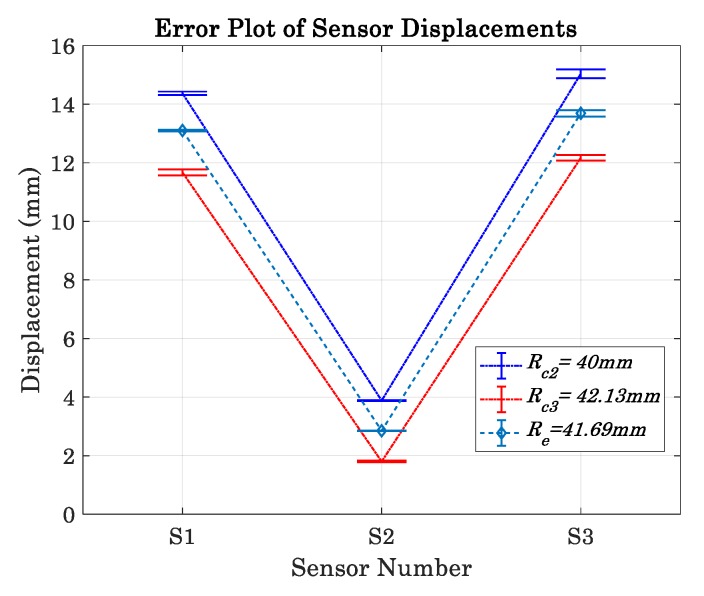
Displacements measurement obtained for the calibration master and an inspected part. Each displacement was measured three times.

**Table 1 sensors-19-04679-t001:** Parametric simulation of the displacements between sensor coordinates and the calibration master.

$R_{c2}=38.00\;\mathrm{mm}$		$R_{c3}=39.00\;\mathrm{mm}$	
Sensor	Simulated	Sensor	Simulated
$y_{12}$	3.33939 mm	$y_{13}$	2.30385 mm
$y_{22}$	2.00000 mm	$y_{23}$	1.0000 mm
$y_{32}$	3.33939 mm	$y_{23}$	2.30385 mm

**Table 2 sensors-19-04679-t002:** TPFK algorithm based on the mathematical simulation: solution of the *S*$\left\{x_{2o},x_{3o},y_{2o},\;y_{3o},a_{c},b_{c}\right\}$ obtained for several initial estimations.

Test	Initial Guess $[x_{2o},x_{3o},y_{2o},\;y_{3o},$ $a_{c},b_{c}]$	# Iterations to Convergence	Ers $1\;\times \;10^{-10}\mathrm{mm}$	$x_{2o}\;\mathrm{mm}$	$x_{3o}\;\mathrm{mm}$	$y_{2o}\;\mathrm{mm}$	$y_{3o}\;\mathrm{mm}$	$a_{c}\;\mathrm{mm}$	$b_{c}\;\mathrm{mm}$
True solution is obtained from the parametric equations:				10.000	20.000	0.000	0.000	10.000	40.000
**Trial 1**	[8,19,1,1,9,39]	23	0.000534	9.998	19.998	0.000	0.000	9.999	40.000
**Trial 2**	[12,18,2,1,4,35]	95 (stopped)	538.63	19.998	20.000	0.001	0.000	9.999	40.000
**Trial 3**	[8,13,2,3,12,43]	95 (stopped)	342.70	10.195	20.000	0.001	0.000	9.999	40.000
**Trial 4**	[15,25,2,2,15,50]	52	10.085	10.017	19.998	0.000	0.000	9.999	40.000
**Trial 5**	[20,25,2,2,20,50]	16	0.000538	10.000	19.998	0.000	0.000	9.999	40.000
**Trial 6**	[20,25,2,2,5,5]	117 (stopped)	0.56 × 10^10^	16.977	40.307	64.830	−9.854	27.854	29.399
**Trial 7**	[1,1,1,1,1,1]	106 (stopped)	2.05 × 10^10^	9.5110	22.302	−61.161	10.345	26.507	−25.103
**Trial 8**	[15,25,−2,−2,15,50]	15	0.000865	10.000	19.998	0.000	0.000	9.999	40.000
**Trial 9**	[20,25,−2,−2,20,50]	45	0.000141	9.999	19.998	0.000	0.000	9.999	40.000
**Trial 10**	[8,19,1,−1,9,39]	90 (stopped)	526 × 10^13^	12.097	26.698	74.837	−1.913	15.937	37.914

**Table 3 sensors-19-04679-t003:** TPFK algorithm based on mathematical simulation: computation of the {*a*, *b*, *R*_3*p*_} of the inspected part by using true and computed $S_{p}$ values.

True Radius *R*	True Center of Inspected Part (a,b)=(10,40) mm								
				Calculation Based on True Values *S_p_* = {10,20,0,0}			Calculation Based on Trial 5 in Table 2 *S_p_* = {10,19.998,0,0}		
	$y_{1}$	$y_{2}$	$y_{3}$	a	b	$R_{3p}$	a	b	$R_{3p}$
38.000	Values from Table 1			10.0000	40.0108	38.0108	9.999	40.0033	38.0033
39.000	Values from Table 1			10.0000	39.9956	38.9956	9.999	39.9879	38.9879

**Table 4 sensors-19-04679-t004:** TPIK algorithm based on CAD simulation: displacements measured between sensor coordinates and the calibration master for two radii by using direct CAD dimensioning tools rounded to *n* = 6 decimal places.

$R_{c2}=38.000000\;\mathrm{mm}$		$R_{c3}=39.000000\;\mathrm{mm}$	
Sensor	Measurement (mm)	Sensor	Measurement (mm)
$y_{12}$	2.130124	$y_{13}$	1.094501
$y_{22}$	1.963025	$y_{23}$	0.963052
$y_{32}$	2.138668	$y_{23}$	1.103195

**Table 5 sensors-19-04679-t005:** TPIK algorithm based on CAD simulation studied for several initial estimations $\left[x_{2o},x_{3o},y_{2o},\;y_{3o},a,b\right]$, and measurement resolutions, *n.*

Trial #	Initial Guess	*n*	$x_{2o}\;(\mathrm{mm})$	$x_{3o}\;(\mathrm{mm})$	$y_{2o}\;(\mathrm{mm})$	$y_{3o}\;(\mathrm{mm})$	a (mm)	b (mm)	*Ers* 10^−7^ (mm^2^)	# Iterations
-	**True** *		10.000	20.000	−1.175	−0.014	10.010	38.788	-	-
1	[10,20,−2,−1,11,39]	6	10.010	20.000	−1.1750	−0.014	10.010	38.788	3.0458	108 (stopped)
2	[10,20,−2,0,9,39]	6	10.012	20.002	−1.175	−0.014	10.012	38.788	2.1151	107 (stopped)
3	[12,21,2,1,11,38]	6	9.048	17.536	72.354	−3.990	17.060	36.135	50	94 (stopped)
4	[10,20,−1.175,−0.014,10.01,38.788]	6	10.010	20.000	−1.175	−0.014	10.010	38.788	3.045	99 (stopped)
5		5	10.010	20.000	−1.175	−0.014	10.010	38.788	1.2075	85
6		4	10.006	20.000	−1.175	−0.014	10.010	38.788	43.69	97 (stopped)
7		3	9.927	19.987	−1.152	0.027	9.927	38.811	1.1	8
8		2	10.583	21.148	−1.331	−0.010	10.574	38.629	11	54

* *True values are measured directly from CAD model*.

**Table 6 sensors-19-04679-t006:** TPFK algorithm based on CAD simulation is studied for two groups of parts.

Part True *R_e_*	Probes Measurement (mm), *n* = 3			Calculated (mm)			Error (mm)		
	$y_{1}$	$y_{2}$	$y_{3}$	a	b	$R_{3p}$	$\left|a-a_{e}\right|$	$\left|b-b_{e}\right|$	$\left|R-R_{e}\right|$
**Group 1: True part center location is fixed at *(a_e_,b_e_)*** =(10.010, 38.788) mm									
38.000	Values from Table 3, *n* = 3			10.0092	38.7851	37.9971	0.001	0.003	0.003
38.250	1.871	1.713	1.880	10.0092	38.7831	38.2451	0.001	0.005	0.005
38.500	1.612	1.463	1.621	10.0093	38.7845	38.4965	0.001	0.003	0.004
38.750	1.353	1.213	1.362	10.0094	38.7895	38.7515	0.001	0.002	0.002
39.000	Values from Table 3, *n* = 3			10.0113	38.7836	38.9956	0.001	0.004	0.004
39.250	0.836	0.713	0.845	10.0095	38.7809	39.2429	0.001	0.007	0.007
39.500	0.577	0.463	0.586	10.0096	38.7964	39.5084	0.000	0.008	0.008
**Group 2: Part center location is arbitrarily (unrestricted positioning)**									
38.000	8.268	9.303	10.742	5.5273	45.8615	37.9977	-	-	0.002
38.000	2.040	2.963	4.277	5.9495	39.5931	38.0215	-	-	0.022
38.000	2.408	2.244	2.422	10.000	39.0798	38.0108	-	-	0.011
38.000	1.067	0.903	1.081	10.000	37.7388	38.0108	-	-	0.011
38.000	1.540	1.376	1.554	10.000	38.2118	38.0108	-	-	0.011
38.000	1.364	0.660	0.310	11.9549	37.4382	38.0035	-	-	0.004
38.000	0.185	0.120	0.398	9.6352	36.9458	38.0026	-	-	0.003

**Table 7 sensors-19-04679-t007:** Calculation of Roundness for $R_{p}$ = 38/40 mm, $R_{e}=38\;\mathrm{mm}$, and frequency ω=5 rad. The system parameters were set to $\;x_{1o}=0,\;x_{2o}=10.000\;\mathrm{mm},\;x_{3o}=20.000\;\mathrm{mm},\;y_{1o}=0,\;y_{2o}=0\;\mathrm{mm},\;\mathrm{and}\;y_{3o}=0$.

Inc. *i*	θ (°)	True Part Center is Fixed and Rotates about a = 10 mm and b = 20 mm by Sampling Increment of δθ= 20° , and Frequency ω=5 rad.					
		Probes Measurement (mm), *n* = 3			Calculated (mm)		
		$y_{1}$	$y_{2}$	$y_{3}$	$a_{3p,i}$	$b_{3p,i}$	$R_{3p,i}$
1	20	3.366	2.349	2.004	10.14	40.445	37.975
2	40	3.337	2.3440	2.024	9.943	40.337	38.003
3	60	3.314	2.307	1.987	9.883	39.452	38.024
4	80	3.351	2.325	1.981	10.097	39.859	37.989
…	…	…	…	…	…	…	…
12	320	3.354	2.355	2.024	10.041	40.590	37.986
13	340	3.315	2.318	2.004	9.860	39.731	38.023
14	360	3.333	2.310	1.975	10.008	39.519	38.006
	Average				$\bar{a}=9.9994$	$\bar{b}=39.9821$	$R_{a}\;$= 37.9992

**Table 8 sensors-19-04679-t008:** Computation of roundness index using CAD data. The system parameters used were $x_{2o}=10.000\;\mathrm{mm},\;x_{3o}=20.000\;\mathrm{mm},\;y_{2o}=-1.175\;\mathrm{mm},\;\mathrm{and}\;y_{3o}=-0.014\;\mathrm{mm}$.

Δθ	True Part Center is Fixed and Rotated about a = 7.873513 and b = 43.48817					
	Probes Measurement, *n* = 3			Calculated		
	$y_{1}$	$y_{2}$	$y_{3}$	a	b	$R_{3p}$
0	1.357	2.445	3.169	5.4487	52.8828	51.8131
2	1.598	2.622	2.941	5.9349	63.4376	62.1237
4	1.904	2.610	2.619	7.8732	62.9325	61.5343
6	2.118	2.413	2.379	9.3908	51.5730	50.3386
8	2.141	2.122	2.335	9.6553	40.5330	39.5875
10	1.972	1.875	2.488	9.1946	34.3121	33.6218
12	1.697	1.792	2.758	8.4051	32.6860	32.1086
14	1.433	1.906	3.033	7.3904	35.1336	34.5015
16	1.296	2.164	3.198	6.2524	41.9387	41.1208
18	1.357	2.445	3.169	5.4487	52.8828	51.8131
20	1.598	2.622	2.941	5.9349	63.4376	62.1237
-	-	-	-	-	-	-
358	1.296	2.164	3.198	6.2524	41.9387	41.1208
	Average			$\bar{a}=\;$7.7273	$\bar{b}=\;$46.1588	$R_{a}\;$= 45.1944

**Table 9 sensors-19-04679-t009:** Displacements of the calibration master measured at two radii by using displacement sensors with ±0.01 mm accuracy.

	$R_{c2}=40.00\;\mathrm{mm}$		$R_{c3}=42.13\;\mathrm{mm}$	
Sensor	Avg./SD (mm)	Measurement (mm)	Avg./SD (mm)	Measurement (mm)
S_1_	($\bar{y_{12}}$, sd_12_)	(14.37, 0.06)	($\bar{y_{13}}$, sd_13_)	(11.67, 0.10)
S_2_	($\bar{y_{22}}$*,* sd_22_)	(3.88, 0.02)	($\bar{y_{23}}$, sd_13_)	(1.80, 0.03)
S_3_	($\bar{y_{32}}$, sd_32_)	(15.04, 0.15)	($\bar{y_{33}}$, sd_13_)	(12.17, 0.10)

**Table 10 sensors-19-04679-t010:** Displacements of the inspected part whose nominal radius is priory known $R_{e}\;$ = 41.03 mm, measured with an accuracy of ± 0.01 mm.

	$R_{e}=41.03\;\mathrm{mm}$	
Sensor	Avg./SD (mm)	Measurement (mm)
S_1_	($\bar{y_{1}}$, sd_1_)	(13.10, 0.03)
S_2_	($\bar{y_{2}}$*,* sd_2_)	(2.85, 0.01)
S_3_	($\bar{y_{3}}$, sd_3_)	(13.69,0.11)

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
