# Peer review of "Three-Point Inverse and Forward Kinematic Algorithms for Circle Measurement from Distributed Displacement Sensor Network"

_sensors, 2019, doi:10.3390/s19214679_

Round 1

Reviewer 1 Report

It is this reviewer's opinion that this paper would be good if there were not so many grammatical errors. I suggest that the author please have a service, or a colleague that understands English grammar, to proofread and correct the grammatical errors in the document before it is re-submitted. 

As shown below, there are too many lines of text in the Abstract and Introduction that require editing: 

A classical problem in quality control of manufacturing mechanical part is the automatic fitting of an arc center and radius. 

A classical problem in the quality control of manufacturing a mechanical part is the automatic fitting of an arc center and radius. 

Because there are always more parts being fabricated than inspected, the technology of measurement has always lagged behind.

It is unclear why that is the reason why measurement technology has lagged.

This paper introduces new procedures and apparatus for fitting arc from distributed sensors with goal to improve the measurement throughput and reduce measurement errors associated with hardware and fitting algorithm.

This paper introduces new procedures and apparatus for fitting arcs from distributed sensors with goal to improve the measurement throughput and reduce measurement errors associated with hardware and fitting algorithm.

We introduce three-point forward kinematic algorithm (TPFK) – a deterministic closed form solution- to calculate circle radius and its center from three independent measurement systems.

We introduce a three-point forward kinematic algorithm (TPFK) – a deterministic closed form solution- to calculate circle radius and its center from three independent measurement systems.

This technique allows deployment of high accuracy gauge system that in general reduces machine and algorithm errors.

This technique allows the deployment of a high accuracy gauge system that in general reduces machine and algorithm errors.

The effectiveness of this direct fitting is evaluated using mathematical, CAD and experimental data set,

The effectiveness of this direct fitting is evaluated using mathematical, CAD, and experimental data set,

The simulations examine the roundness index in relation to measurement precision, sampling angle, nominal radius and fluctuation of part. 

The simulations examine the roundness index in relation to measurement precision, sampling angle, nominal radius, and fluctuation of the part.

The paper discusses interesting outcomes one of which is that once measurement system is calibrated, parts radii can be inspected fast and accurately regardless of human error associated from loading part into the measurement system; making our methods more fit for inspection of arc parts in mass-production environment.

One of the beneficial outcomes is that once the measurement system is calibrated, the radii of the parts can be inspected quickly and accurately, regardless of human error associated from loading the parts into the measurement system, which makes this method a much better fit for inspection of arc parts in mass-production environment than was is currently employed.

Fitting surface geometry of part is the process of establishing substitute geometry from real data points collected using metrology instruments on real components and applied to optimize a relevant model for the particular feature being measured.

Fitting surface geometry to a part is the process of estimating the interpolated geometry between the real data points collected using metrology instruments on real components. This method is an efficient way to develop a geometric model of the feature being measured.

Perhaps fitting a circular arc  ( say, for a fillet or a section of spherical mirror or cylinder) are much studied problem that has motivated a large amount of literature in science and engineering with application to  microwave measurement (Umbach and Jones, 2003), computer vision(Safaee-Rad et al., 1992), design tolerance(Traband, Medeiros and Chandra, 2004), metrology and inspection of mechanical parts  (Landau, 1987)(Thomas and Chan, 1989), archaeology(Thom, 1955), geodesy(Robinson, 1961) , and many others. 

The problem of fitting a circular arc (say, for a fillet or a section of a spherical mirror or cylinder) has motivated a large amount of literature in science and engineering with applications to microwave measurement (Umbach and Jones, 2003), computer vision (Safaee-Rad et al., 1992), design tolerance (Traband, Medeiros, and Chandra, 2004), metrology and inspection of mechanical parts (Landau, 1987) (Thomas and Chan, 1989), archaeology (Thom, 1955), geodesy (Robinson, 1961), and many others. 

For example, camshaft is one of the most important components in internal combustion engines, and their manufacturing precision significantly correlate to the performance of engine (Lin et al., 2017).

For example, the camshaft is one of the most important components in internal combustion engines, and the manufacturing precision of the camshaft significantly correlates to the performance of engine (Lin et al., 2017).

In recent years, attention has been made to issue that arise in using measurement technologies in automated inspection, particularly the Coordinate Measuring Machines (CMMs) (Traband, Medeiros and Chandra, 2004).

In recent years, attention has been made on the issue that arises when using measurement technologies in automated inspections, particularly with the coordinate measuring machines (CMMs) (Traband, Medeiros and Chandra, 2004).

Such kind of measurement can be affected by machine mechanical errors, part form errors, measurement algorithm applied, and fitting algorithm employed by machine control unit (Hopp, 1993)(Schwenke et al., 2008)(Phillips et al., 2003).

This type of measurement can be affected by machine mechanical errors, part form errors, inaccuracies within the measurement algorithm and the fitting algorithm employed by the machine control unit (Hopp, 1993) (Schwenke et al., 2008) (Phillips et al., 2003).

In fitting best-fit geometry, more points are sampled to reduce the influence of outliers and minimize the effect of any measurement errors.   There are many optimization criteria used to determine the parameters depending on the application, and they vary with the used to determine the parameters depending on the application, and they vary with the statistical error model (Pegna and Guo, 1998).  

The improve geometric fitting, what is typically done is to sample a larger number of points to reduce the influence of outliers and minimize the effect of measurement errors. There are many optimization criteria used to determine the parameters depending on the application, and they vary with the method used to determine the parameters depending on the application, as well as vary with the statistical error model (Pegna and Guo, 1998).

The fitting problem could be classified into algebraic and geometric methods (Rusu et al., 2003)(De Guevara et al., 2011).

The fitting problem could be classified into algebraic and geometric methods (Rusu et al., 2003), (De Guevara et al., 2011).

Examples of geometrical methods include minimizing the mean square error (MSE) sum,  the inversion method of Brandon and Cowley(Brandon and Cowley, 1983), the minimum-circumscribed (MC) and maximum-inscribed (MI) criteria are also popular (Muralikrishnan and Raja, 2008).

Examples of geometrical methods include minimizing the mean square error (MSE) sum, the inversion method of Brandon and Cowley (Brandon and Cowley, 1983), the minimum-circumscribed (MC) and maximum-inscribed (MI) criteria are also popular (Muralikrishnan and Raja, 2008).

Perhaps the mostly used one is the algebraic fitting that minimize the sum of squared deviations, or primarily known by the least square (LS) methods, which are robust against outlier and easy to implement. 

The most widely used method is the algebraic fitting, which minimizes the sum of squared deviations (i.e. least square (LS) fitting), because the method is robust against outlier and it is easy to implement. 

Example of robust method used in extraction of circle is Hough transform which is mainly used in digital image (Rusu et al., 2003), full LS, reduced LS and modified LS  (Umbach and Jones, 2003), Delogne-Kasa for noisy data (Zelniker and Clarkson, 2006).

An example of a robust method used in the extraction of a circle is the Hough transform, which is mainly used in digital imaging (Rusu et al., 2003), full LS, reduced LS, modified LS (Umbach and Jones, 2003), and Delogne-Kasa for noisy data (Zelniker and Clarkson, 2006).

When fitting circle, the LS methods results in non-linear and implicit equations  that can only be solved numerically, often following linearization techniques (Landau, 1987).

When a fitting circle, the LS methods result in non-linear and implicit equations that can only be solved numerically, often following linearization techniques (Landau, 1987).

While the previous aforementioned statistical fitting relies on availability of more data points than the unknown parameters of the substitute geometry model, however, one motivation of this paper is that in mass production environment the quality of part geometry in often scenarios should be checked quickly for economic and practical reasons, therefore there must be exactly as many data points collected as required.

Since the above statistical fitting methods rely on the availability of more data points than the unknown parameters of the substitute geometry model, the methods are not amenable to the high flow rate of mass production. Therefore, one of the motivations of this paper is that in a mass production environment, the quality of part geometry in most scenarios should be checked quickly for economic and practical reasons. There is a need for an accurate method that can utilize a reduced set of data points.

Few related studies are available today, for example the National Institute of Standards and Technology (NIST) found that the Three-Point measurement procedures and its  task uncertainties depend only on the mean and variance of point measurement errors and are essentially independent of their statistical distribution (Hopp, 1994). 

Few related studies are available today, for example the National Institute of Standards and Technology (NIST) found that the three-point measurement procedures depend only on the mean and variance of point measurement errors and are essentially independent of their statistical distribution (Hopp, 1994). 

Although CMM provides accurate reconstruction of complex shape from scattered points, however conventional CMM measurement is slow and may not meet the production throughput requirement.

Although CMM provides an accurate reconstruction of complex shape from scattered points, conventional CMM measurement is slow and may not meet the production throughput requirement.

This is mainly because CMM system carry out the measurement serially from one point to another by sensor attached to robot arm’s end effector in a global coordinate system.

This is mainly because the CMM system carries out the measurement serially from one point to another by a sensor attached to the robot arm’s end effector in a global coordinate system.

This type of sensing technology depends on the complex machine form making it subject to errors.   The purpose of this paper is to develop an efficient and automatic method and apparatus for fitting the radius of parts in production line by measuring three-point from displacement sensors located at independent coordinate systems.

This complexity involved in this type of sensing technology is subject to errors. The purpose of this paper is to develop an efficient and automatic method and apparatus for fitting the radius of parts in production line by measuring the three-point from displacement sensors located within an independent coordinate system.

The paper is organized as follow: section 2.1 derives a closed-from solution algorithm for calculation of circle and radius from data points measured by three independent Cartesian coordinate systems.

The paper is organized as follows: Section 2.1 derives a closed-form solution algorithm for calculation of circle and radius from data points measured by three independent Cartesian coordinate systems.

Extensive numerical simulation studies are conducted to validate proposed method by using mathematical model in sections 3.1,3.2&3.5, and CAD simulation in sections 3.3,3.4&3.6.

Extensive numerical simulation studies are conducted to validate the proposed method by using a mathematical model in Sections 3.1, 3.2, and 3.5, and CAD simulations in Sections 3.3, 3.4, and 3.6.

Finally, section 3.7 describe a simplified experiment and validate the proposed methods.

Finally, Section 3.7 describes a simplified experiment that validates the proposed methods.

Solidworks

SolidWorks

Mathworks

MathWorks

Author Response

Please see attached Document and revised Manuscript.

Reviewer 2 Report

The details of both of TPIK and TPFK are not interesting and boring.

However, the Calibration-Master you introduced is interesting and the experiment and simulation of the Calibration-Master is effective.

Author Response

(The authors gave the same response as above.)

Reviewer 3 Report

The research paper presents a novel idea for circle fitting using TPFK and TPIK algorithms. The author has done a good job in elaborating the topic especially in terms of the background, technology gap and the proposed method. Here are a few insights on things that should be addressed : -

Typos on page 3 "closed-from solutions" and page 10 "priory". If EACH inspection part comes with its own calibration master would'nt it make it  more difficult to accept this procedure at a large scale production.  The measurement part needs to be rotated more than once to ensure multiple readings and repeatability of the process. What is the resolution/sensitivity of the displacement sensors? 50% of the initial guesses either did not converge or stopped in between. Seems contrary to how this can reduce the measurement time and boast for a reliable production method. There should be a reference sensor on the fixture whose sole purpose is to monitor the drifts in the offset due to mechanical vibration, temperature fluctuations and noise. This sensor should not make contact with the calibration master. This way the effect of external parameters on the performance of the sensors can be negated during computation. How does one make sure that the stiffness of the spring loaded sensors is identical? Since the setup requires 3 of them and any difference in stiffness will affect the measurements.

Author Response

(The authors gave the same response as above.)

Reviewer 4 Report

The author touched the important area of the roundness measurement. Nevertheless, from the engineering point of view, there are many things missing.

When building a sensor, the 3 major factor comes to play: accuracy, repeatability, and precision. This paper is not presenting it. Although the mathematical models seem to be correct, the experiment cannot be based on simple measurement. It must be supported by engineering assessment of those 3. Without it, the industry won't be interested in using this sensor in practice. And the author mentioned about industrial problems in the first sentence of the abstract. The paper is focused on modeling and models validation, the experiment is added by chance.

Please, consider adding missing parts of statistical evaluation of the sensor and send paper again.

Author Response

(The authors gave the same response as above.)

Round 2

Reviewer 4 Report

I recommend to publish the paper without changes.